# Development and Validation of the Teacher Career-Related Support Self-Efficacy (TCSSE) Questionnaire

**DOI:** 10.3390/bs13010036

**Published:** 2022-12-31

**Authors:** Anna Parola, Marina Pettignano, Jenny Marcionetti

**Affiliations:** 1Department of Humanities, University of Naples Federico II, 80133 Naples, Italy; 2Department of Education and Learning, University of Applied Sciences and Arts of Southern Switzerland, 6928 Manno, Switzerland

**Keywords:** career-related teacher support self-efficacy, career-related teacher support, adolescents, career transition

## Abstract

Background: Career-related teacher support self-efficacy refers to the teacher’s confidence in his/her own ability to support students’ career choices. To our knowledge, there are neither studies that focus on this topic nor instruments to measure it. Therefore, the current study aimed at developing and validating an instrument to assess teacher career-related support self-efficacy (TCSSE). Methods: In a multi-step process, items were developed and three studies that involved Italian in-service teachers were conducted. In Study 1 (*n* = 232), the TCSSE was developed and exploratory factor analysis on the 51 initial items was performed. In Study 2 (*n* = 294), six factors—Get Ready, Empower Self, Get Curious, Empower Skills, Emotional Support and Instrumental Support (α = 0.885)—across 36 items were tested through confirmatory factor analysis (CFA). In Study 3 (*n* = 100), the reliability of TCSSE was tested. Results: The EFA performed in Study 1 suggested a six-factor solution with 36 items. The results of Study 2 confirmed the six-factor structure (χ^2^ (579) = 1387.965, *p* < 0.001, CFI = 0.964, TLI = 0.961, RMSEA = 0.069), the internal consistency (α = 0.863 for Get Ready, α = 0.857 for Empower Self; α = 0.864 for Get Curious; α = 0.909 for Empower Skills; α = 0.881 for Emotional Support; α = 0.885 for Instrumental Support) and validity of the TCSSE. Finally, in Study 3, the reliability of TCSSE was also confirmed. Conclusion: The TCSSE questionnaire can provide researchers and practitioners with a new and reliable measure to assess teacher career-related support self-efficacy. Suggestions for future studies and practice are also provided.

## 1. Introduction

During the life span, individuals are required to manage several career transitions, such as the transition between educational systems (for example the transition from middle school to high school), school-to-work transitions and career changes. Adolescents and young people are finding it increasingly difficult to manage these transitions due to unstable and challenging environments. Specifically, career pathways unfold in an unpredictable environment characterized by the following three major challenges: technological evolution and digitalization, economic recession and labor market issues, and environmental factors. These issues make it difficult to construct sustainable careers, find decent work and build a decent life [1]. Indeed, what is traditionally learned in school will not necessarily be related to future job opportunities [2]. New skills, competencies and qualifications are required to master career transitions and schools should promote the enhancement of abilities that can be used by adolescents to better orient themselves in education and, then, in the labor market. In a recent study by OECD and WorldSkills [3] on education and employers, young people were asked about their levels of awareness about the changes affecting the new world of work and how confident they felt about their ability to negotiate them. Regarding school, 56% of young people knew what they wanted to do for work in the future, however, they did not feel supported by their education system, and 44% feared that their skills or knowledge would not be in demand in the future.

During basic and compulsory education, the development of skills such as coping with and overcoming problems, barriers and setbacks, as well as promoting a sense of curiosity and exploration, together with a positive attitude towards career development, have become central to adolescents’ career guidance [4,5]. Schools should be conceived as an important training period for career development and future planning. In addition to targeted orientation activities promoted by qualified staff, teachers are important actors in fostering good career transitions. 

Career-related teacher support is generally defined as anything a teacher does that can facilitate the career planning of students. Specifically, it refers to the teachers’ behavior toward students as invested caregivers and a source of positive career expectations, information and support in career development [6]. According to the career construction theory (CCT) [7,8], individuals shape their sense of self, which includes one’s career self-image, and balanced positive and negative experiences encountered. In this paradigm, teacher support is conceived as the top of the students’ positive experiences [9,10]. Metheny and colleagues [11] identified the following four factors in career-related teacher support: invested effort, positive regard, positive expectations, and accessibility. Invested effort reveals the will to behave in order to support future students’ success. Positive regard refers to teachers being emotionally connected to their students and caring for students’ needs. Expectations refers to teachers communicating their positive expectations of students’ future educational and vocational success. Finally, accessibility relates to teachers being perceived by students as helpful in attending to their needs when they desire information or support. In a recent review, Wong and colleagues [6] found the following four main roles that correspond to these traits: teachers as invested caregivers, teachers’ role in fostering students’ self-efficacy, teachers as a source of positive career expectations, and teachers as resource persons available for support and information.

Studies have showed a strong link between teachers’ perceived interest in students and their career commitment [12]. Moreover, increased support reduced students’ perceptions of barriers in their own educational and professional development [12]. Related to the role of teachers in fostering students’ self-efficacy, studies have revealed the positive influence of teachers on attitudes, motivation, and students’ self-confidence [13] and in the identification of their strengths and vocational interests [14,15]. Finally, teachers can give instrumental support such as advice, direct guidance, and information [12] that can assist in making career choices [16].

In the literature, several self-report measures of perceived teacher support by children or adolescents exist. Among others, the Teacher Support Scale (TSS) has become one of the most widely used questionnaires [11] that assesses students’ perception of the four traits of teacher support mentioned above. However, to our knowledge, there are no specific studies that focus on teachers’ perception of their role of support in students’ career development, and no instruments exist to measure it. This type of measure of one’s role in the career choice process has been developed, but only for parental support. In fact, the My Children’s Future Scale (MCFS) [17] measures the support provided by parents for their children’s careers. The scale was included in a study that aimed to assess the career support that parents themselves perceived to provide and the support perceived by adolescents [18]. It is indeed important to measure both points of view since, for example, adolescents who receive support may have different views on their parents’ supportive behavior [13]. Moreover, Kenny and colleagues [19] showed that teenagers who perceive adults as supportive have more consideration for work importance and have a better view of their ability to pursue the career to which they aspire. Since the main adult references for adolescents are parents and teachers, it is worth considering both actors in their role in career support. In particular, teachers may also provide social support directly through career interventions [20]. Teachers, in addition to giving general support, may be called upon to use specific career development methods. For example, Ozdemir and Aydın [21] analyzed the effect of brief motivational interviewing (MI) training on middle school teachers’ sense of efficacy for student engagement (especially for students with difficult socioeconomic conditions). The training was effective in improving teachers’ perceived efficacy and enabled teachers to stimulate reflection on what career support implies. Teachers reported that they recognized making some mistakes when offering career support, “such as not respecting their students’ autonomy, not understanding their students’ values and context” ([21], p. 275).

Self-efficacy refers to people’s estimated probability of success in a particular task [22]. It determines how much effort is expended on a specific activity and the perseverance in maintaining this effort [23] and the actual performance [22]. Career-related teacher support self-efficacy can be defined as the teacher’s confidence in his/her own ability to support students’ career choices. In other words, it is conceived as the teachers’ perceived estimated probability of success in different tasks related to students’ career support. On the other hand, teachers’ self-efficacy is related to being more engaged in supportive relationships with students [24], and it can be inferred that teachers’ perceived self-efficacy in different tasks related to career support is related to teachers’ implementation of these activities. Consequently, improving teachers’ perceived self-efficacy in these tasks should lead them to implement them more frequently and effectively [25].

Still, despite the importance of establishing teachers’ career-related support perceived efficacy, there is little research that focuses on teacher perceived efficacy in this role. A possible explanation for this paucity of research is certainly that no measure of teachers’ perceived self-efficacy in providing career support has yet been developed and tested. To address these limitations, the present study sought to develop a new psychometrically sound instrument for teachers’ career-related support perceived self-efficacy.

Having an assessment measure of teachers’ career-related support perceived self-efficacy has important theoretical and practical implications. Specifically, it will allow (a) enhanced theoretical knowledge of teachers’ perceptions regarding their ability to provide this specific type of support, (b) enabled research of the relationship between teachers’ self-efficacy in this task and possible antecedents and outcomes, and (c) provision of an important self-report measure for assessing the effectiveness of specific training aimed at improving the support provided by teachers for career transitions of their students. Enhancing teacher self-efficacy will help support behaviors by producing a snowball effect on students and their career choices. Hence, the main aim of this study was to develop an instrument to measure the construct of teacher career-related support self-efficacy (TCSSE). For the development of this new scale, we followed the recommendations of Boateng and colleagues [26]. The three phases of creating a scale—item development, scale development, and scale evaluation—were performed.

## 2. Study I: Developing a List of Scale Items and Extraction of Latent Factors

The aim of Study 1 was to develop preliminary items for measuring TCSSE and to test the factor structure of the measure. 

In the first phase, we created the initial set of items. We followed the following two steps: (i) item generation through literature review (see the Introduction section) and assessment of existing scales and indicators of those domains of teacher career-related support; (ii) content validity, which refers to the adequacy with which a measure assesses the domain, assessed through evaluation by five expert judges, such as researchers that are potential users of the scale. In step 1, we chose to determine the domains (and possible dimensions of these domains) a posteriori. Despite there being an established framework (CCT; [7]) guiding the study (which allowed the generation of some of the items included in the scale), no previous instruments that assess the teacher career-related support self-efficacy existed. Thus, the set of potential items was developed after having identified the broad area that should be assessed in the new measure, i.e., the teacher career-related support self-efficacy. For item development, we followed a recommended multi-step procedure to ensure high item validity [27]. Firstly, three experts identified the specific scales referring to the abilities and skills needed for career transitions (i.e., career adaptability, [28]; 21st century skills, [29]) and related to adult career-related external support measurement (parental support administered to children, [30]; teacher support administered to pupils, [10]; parental support administered to parents, [18]). We then used a deductive item-generating strategy [27,31] by either developing new items or adapting items from existing scales. All items generated by the experts underwent an internal content validity review; experts provided comments for each item, and these were amended when necessary. In step 2, five experts in the field of psychology who have a PhD degree and have conducted research related to scale development evaluated content validity for the initial items. The external experts were blind to the aim of the process. Taking into account the target population, experts were asked to assess the extent to which the item measured the intended reaction/domain. An agreement level higher than 80% between experts was considered adequate to retain each of the items. This process resulted in a set of 51 items.

In the second phase, items had to be turned into a measuring construct. The following four steps were followed: (i) pre-testing involving 20 teachers helped to ensure that items were understandable and meaningful to the teacher population, during which no problematic questions were pointed out by them; (ii) item administration to 232 teachers was performed, which permitted (iii) item reduction and (iv) extraction of latent factors. Steps (ii), (iii), and (iv) are described in the following sections.

### 2.1. Materials and Methods

#### 2.1.1. Participants and Procedure

A sample of 232 in-service Italian teachers was enrolled to test for factor structure. Teachers were aged 31–56 (M = 37.25; SD = 6.40); 29% were males and 71% were females. The majority of the sample had teaching seniority of more than ten years (67%). Teachers were recruited during courses for in-service teachers and were asked to fill out the questionnaire including preliminary items during the refresher courses. Inclusion criteria were as follows: (a) being a native Italian speaker; (b) being an in-service teacher in an Italian school; (c) providing informed consent. Exclusion criteria were as follows: (a) illiteracy; (b) inability to complete the assessment due to vision impairment; (c) being a pre-service teacher.

Approval from the University Research Ethics Committee was obtained for collecting data. Participants were informed about a complete guarantee of confidentiality, the voluntary nature of participation, and their right to discontinue filling in the questionnaire at any point.

#### 2.1.2. Measures

The participants completed all 51 items of the TCSSE measure. Respondents were required to evaluate the extent to which each item described their perceived self-efficacy in the specific tasks proposed on a five-point scale ranging from 1 (= not strong) to 5 (= strongest).

#### 2.1.3. Data Analysis

A preliminary inspection of the item performance was completed. Before moving into the more restrictive confirmatory factor analysis, an exploratory factor analysis (EFA) was performed to evaluate a possible measurement structure. The weight least square adjusted for mean and variance (WLSMV) estimator was used. To evaluate the adequacy of models to the data, the Chi-square statistic, the comparative fit index (CFI), and the root mean square error of approximation (RMSEA) with associated 90% confidence intervals were used. The following cut-off criteria were chosen to evaluate the goodness of fit: (a) statistical non-significance of the χ^2^; (b) an RMSEA lower than 0.08; (c) a CFI and TLI higher than 0.95; (d) an RMSEA and SRMR lower than 0.08 [32,33,34,35]. No missing data were found in the dataset. Then inter-item correlations were calculated to examine the extent to which scores on one item were related to scores on all other items in a scale. All the analyses were performed with Mplus 8 [36].

### 2.2. Results

Exploratory Factor Analysis (EFA) was conducted on the 51 items with the WLSMV estimator and geomin rotated solution. The analysis suggested a six-factor solution. No factor loadings lower than 0.45 emerged. The model fit was adequate (χ^2^ (984) = 1532.305; *p* < 0.001), RMSEA = 0.049; 90% CI (0.044–0.054), CFI = 0.988; TLI = 0.984; SRMR = 0.031. In this phase, items with cross-loadings across factors that exceeded 0.30 were deleted. Moreover, items producing inter-item correlations that exceeded r ≥ 0.80 were deleted. A total of 36 items were retained in the final version of the scale.

## 3. Study 2: Scale Evaluation

Scale evaluation refers to testing the previous model through the assessment of (i) dimensionality, (ii) reliability, and (iii) validity. The first aim of the second study was to confirm the six-factor structure of the items selected in Study 1 in a new sample of teachers. We expected the six-factor structure to fit the data well. The second aim was to assess the concurrent validity between teacher career-related support self-efficacy and the following two other self-efficacy constructs: career decision self-efficacy and teacher self-efficacy. We expected a positive correlation between all the dimensions of these constructs.

### 3.1. Materials and Methods

#### 3.1.1. Participants and Procedure

A sample of 294 in-service Italian teachers served to confirm the adequacy of the six-factor solution and the validity of the measure. Teachers were aged 26–58 (M = 39.55; SD = 7.81); 21.8% were males and 78.2% were females. The majority of the sample had teaching seniority of more than ten years (71%). Teachers were recruited during courses for in-service teachers. They were asked to fill in the questionnaire during refresher courses. Inclusion criteria were as follows: (a) being a native Italian speaker; (b) being an in-service teacher in an Italian school; (c) providing informed consent. Exclusion criteria were as follows: (a) illiteracy; (b) inability to complete the assessment due to vision impairment; (c) being a pre-service teacher. Approval from the University Research Ethics Committee was obtained for collecting data. Participants were informed about a complete guarantee of confidentiality, the voluntary nature of participation and their right to discontinue filling in the questionnaire at any point.

#### 3.1.2. Measures

##### Teacher Career-Related Support Self-Efficacy

A questionnaire including 36 items was administered. Respondents were required to evaluate the extent to which each item described their perceived self-efficacy in specific support tasks on a five-point scale ranging from 1 (= not strong) to 5 (= strongest). The introductory sentence read “How capable do you feel you are of supporting students in...” and is followed by the specific activity.

See Appendix A (Table A1) for the Italian version (validated in this paper) and English translation of the TCSSE. 

##### Career Decision Self-Efficacy

Career decision self-efficacy was measured with the Italian Short version of the Career Decision Self-Efficacy Scale [37,38]. The scale includes 25 items rated on a 5-point Likert scale ranging from 1 (= not at all confident) to 5 (= totally confident). This instrument includes the following five dimensions: Self-appraisal (5-item, e.g., “Accurately assess your abilities”), Occupational information (5-items, e.g., “Find out the employment trends for an occupation over the next 10 years”), Goal selection (5-items, “Choose a career that will fit your preferred lifestyle”), Planning (5-items, “Make a plan of your goals for the next 5 years”), and Problem solving (5-items, e.g., “Change occupations if you are not satisfied with the one you enter”). The dimensions are defined as the sum of the items belonging to them (range 1–5). Higher scores indicate higher levels of career decision self-efficacy. Cronbach’s α were 0.73 (Self-appraisal), 0.78 (Occupational information), 0.83 (Goal selection), 0.69 (Planning), and 0.75 (Problem solving) for the original version, and 0.67 (Self-appraisal), 0.58 (Occupational information), 0.64 (Goal selection), 0.69 (Planning), and 0.64 (Problem solving) for the Italian version.

##### Teacher Self-Efficacy

Teacher self-efficacy was measured with the Italian version of the Teacher Self-Efficacy Scale [39,40]. The scale includes 10 items rated on a 4-point Likert scale ranging from 1 (= not true at all) to 4 (= totally true). The dimensions are defined as the sum of the items belonging to them (range 1–5). Items examples are “I am confident in my ability to be responsive to my students’ needs even if I am having a bad day” and “If I try hard enough, I know that I can exert a positive influence on both the personal and academic development of my students”. Cronbach’s α was between 0.76 and 0.82 for the original version (validated on three samples of teachers) and 0.86 for the Italian version.

#### 3.1.3. Data Analysis

Based on the results of the previous EFA (see Study 1), a six-factor solution with 36 out of the 51 original items was tested through CFA. The WLSMV estimator was used. To evaluate the adequacy of models to the data, the Chi-square statistic, the CFI, and the RMSEA with associated 90% confidence intervals were used. The following cut-off criteria were chosen to evaluate the goodness of fit: (a) statistical non-significance of the χ^2^; (b) an RMSEA lower than 0.08; (c) a CFI and TLI higher than 0.95; (d) an RMSEA lower than 0.08 [32,33,34,35]. No missing data were found in the dataset. The internal consistencies of factors were evaluated by computing Cronbach’s alpha (α). Concurrent validity was tested through correlation analysis of teacher career-related support self-efficacy with career decision self-efficacy and teacher self-efficacy and interpreted using the following classical benchmarks: *r* < 0.10, trivial; *r* from 0.10 to 0.30, small; *r* from 0.30 to 0.50, moderate; *r* > 0.50, large. All the analyses were performed with Mplus 8 [36].

### 3.2. Results

#### 3.2.1. Confirmatory Factor Analysis

Based on the previous EFA, a six-factor solution was tested through confirmatory factor analysis on the third sample. The model provided an adequate fit to the data, (χ^2^ (579) = 1387.965, *p* < 0.001, CFI = 0.964, TLI = 0.961, RMSEA = 0.069 (90% CI: 0.064, 0.074)). The six factors are Get Ready (5 items), Empower Self (5 items), Get Curious (5 items), Empower Skills (9 items), Emotional Support (5 items), and Instrumental Support (7 items). Get Ready refers to support given in order to make aware students that they need to prepare for a career choice. An example item was “Make them aware of the importance of preparing for their professional future”. Empower Self refers to the teachers’ role in encouraging students to make decisions by themselves and take responsibility for their actions. An example item was “Doing what’s right for them”. Get Curious refers to the teachers’ role in promoting career exploration. An example item was “Observing different ways of doing things”. Empower Skills refers to the role of teachers in cultivating and fostering students’ enhancement of life skills. An example item was “Being optimistic”. Finally, the two support dimensions refer to the role of teachers in providing guidance. Emotional Support explores teachers’ perceived efficacy to support learners’ autonomous choices by making them aware of their own resources, while Instrumental Support assesses teachers’ perceived efficacy to provide specific career information, as well as opportunities to talk about one’s interests, abilities and dreams and to find useful information to make informed choices. Items examples were “Encourage them to consider their abilities and strengths when thinking about what to do in the future” (Emotional Support), and “Encourage them to talk about their desires and hopes for their professional future” (Instrumental Support).

As reported in Table 1, all standardized factor loadings were statistically significant and ranged from .653 (item 32; Instrumental Support) to .938 (item 28; Emotional Support).

#### 3.2.2. Internal Consistency and Validity

Cronbach’s alpha revealed that TCSSE has good internal consistency in each domain: α = 0.863 for Get Ready, α = 0.857 for Empower Self; α = 0.864 for Get Curious; α = 0.909 for Empower Skills; α = 0.881 for Emotional Support; α = 0.885 for Instrumental Support. Correlations among latent factors are displayed in Table 2. Moderate-to-large correlations among the six domains were found. Moderate correlations were found between Get Ready and Empower Self (*r* = 0.432), between Empower Self and, respectively, Get Curious (*r* = 0.496), Empower Skills (*r* = 0.440), Emotional Support (*r* = 0.465) and Instrumental Support (*r* = 0.461). Large correlations were found between Get Ready and, respectively, Get Curious (*r* = 0.644), Empower Skills (*r* = 0.552), Emotional Support (*r* = 0.596), and Instrumental Support (*r* = 0.566); between Get Curious and Empower Skills (*r* = 0.601), Emotional Support (*r* = 0.656), and Instrumental Support (*r* = 0.641); and between Empower Skills and, respectively, Emotional Support (*r* = 0.610) and Instrumental Support (*r* = 0.697). Finally, a large correlation between Emotional Support and Instrumental Support was also found (*r* = 0.665).

Validity analyses are displayed in Table 3. Specifically, the associations between TCSSE dimensions and the hypothesized validity measures were significant and in the expected direction. Moderate-to-large correlations among career decision self-efficacy, i.e., Self-appraisal, Occupational information, Goal selection, Planning and Problem Solving with the six dimensions of TCSSE emerged. Moreover, moderate correlations between teacher self-efficacy and Get Ready, Empower Self, Empower Skills, and Instrumental Support and large correlations between teacher self-efficacy and Get Curious and Emotional Support were also shown. 

## 4. Study 3: Test-Retest Reliability

The aim of Study 3 was to assess the stability of the TCSSE using test-retest reliability methods. Test-retest reliability was indicated by the intraclass correlation coefficient (ICC), which was calculated through two repeated measures of the TCSSE, the second performed 4 weeks after the first one.

### 4.1. Materials and Methods

#### 4.1.1. Participants and Procedure

A sample of 125 in-service teachers who were taking a refresher course filled in the questionnaire at Time 1 during the class. Inclusion criteria were as follows: (a) being a native Italian speaker; (b) being an in-service teacher in an Italian school; (c) providing informed consent. Exclusion criteria were as follows: (a) illiteracy; (b) inability to complete the assessment due to vision impairment; (c) being a pre-service teacher. Teachers were asked to complete the same questionnaire as a web-based survey after 4 weeks. A total of 25 teachers did not fill in the online questionnaire. Of the 100 teachers who completed the questionnaire at both Time 1 and Time 2, 32.7% were males and 67.3% were females (M = 38.6; SD = 7.10). The majority of the sample had teaching seniority of 5–10 years (54%).

#### 4.1.2. Measures

The participants completed the TCSSE questionnaire. Respondents were required to evaluate the extent to which each item described their perceived self-efficacy in the specific tasks proposed on a five-point scale ranging from 1 (= not strong) to 5 (= strongest).

#### 4.1.3. Data Analysis

Test-retest reliability of each scale was estimated using the two-way mixed ICC [41,42,43,44,45]. ICC values between 0.5 and 0.75 indicate moderate reliability, values between 0.75 and 0.9 indicate good reliability and values greater than 0.90 indicate excellent reliability [45].

### 4.2. Results

Test-retest reliability showed satisfying results, as follows: the two-way mixed ICC was equal to 0.763, 95%CI (0.648, 0.841) for the Get Ready scale, to 0.668, 95%CI (0.506, 0.776) for the Empower Self scale, to 0.811, 95%CI (0.720, 0.783) for the Get Curious scale, to 0.846, 95%CI (0.771, 0.896) for the Empower Skills scale, to 0.792, 95%CI (0.691, 0.860) for the Emotional Support scale, and 0.840, 95%CI (0.763, 893) for the Instrumental Support scale. These findings show that the TCSSE is a reliable measure.

## 5. Discussion

Despite the growing interest in the role of teachers to support the career development of students, no measure that assesses the teachers’ perception of their efficacy in giving career support exists in the literature. This study aimed to fill this research gap by developing a new measure of teacher career-related support self-efficacy.

In Study 1, 51 potential items to measure TCSSE were generated through a literature review and assessment of existing scales. Considering the cross-loadings across factors and the inter-item correlations, a total of 36 items were retained in the final version of the scale. The six-factor structure (Get Ready, Empower Self, Get Curious, Empower Skills, Emotional Support and Instrumental Support) was confirmed in Study 2. Good internal consistency was found for each dimension. Moreover, the TCSSE dimensions were moderately correlated with each other, supporting the appropriateness of considering them as dimensions of a single construct, but distinct from each other. Validity analyses were also performed through inspection of correlations between TCSSE dimensions, career decision self-efficacy [37,38] and teacher self-efficacy [39,40]. Significant positive correlations were found between the TCSSE measure and the well-consolidated measures of career decision self-efficacy and teacher self-efficacy, supporting the validity of the TCSSE. Results suggest that the more teachers feel self-efficacious (as teachers), the more they will have those core competences that will enable them to also feel efficacious in properly supporting students in their career choices. Furthermore, the more teachers feel self-efficacious in making their career choices, the more confident they will feel in establishing themselves as a model, giving information, and maintaining a positive attitude, which they will also convey to students, toward their task of choosing education/training. Finally, the 4 week test-retest reliability in Study 3 also provided satisfying results. 

The findings confirmed that the TCSSE is a valid and reliable instrument for assessing teacher career-related self-efficacy. The six dimensions also cover well the different activities that teachers should perform to support their students’ career choices. First of all, it explores teachers’ self-efficacy in helping students develop those skills, or career adaptabilities, needed to cope with transitions in today’s context [7,28]. This helps them to understand the importance of getting active and preparing to make a choice (Get Ready), in promoting the exploration of professions and the self (Get Curious), in encouraging students to believe in their own possibilities and ability to make decisions and take responsibility (Empower Self). Added to these is the teacher’s ability to foster life skills development in students (Empower Skills), i.e., those transferable skills and competences such as critical sense and creativity that enable them to maintain a positive attitude [46,47]. Finally, two dimensions relating to the ability to give Emotional and Instrumental Support were considered. The first is fundamental, since it allows the teacher to make the student aware of his or her own resources and capabilities and to make him or her more confident in him/herself and autonomous in his/her choices. Hence, it implies that teachers promote reflection about students’ own abilities and resources, i.e., a positive self-evaluation which maintains motivation to pursue the task [48] and also permits to plan future steps for self-development. The second is also important, since it allows the teacher to provide useful information and create opportunities favorable to the development of the skills necessary to make informed choices. This permits the teacher to provide a role model for students to follow, also showing them how information can be found and critically used.

However, the studies performed to create and validate the TCSSE also have some limitations. Firstly, all of the teachers participating came from higher secondary schools. Although the measures developed and validated for teachers are often general and do not refer to specific education but to their role as a teacher, in order to generalize the measure, it could be interesting to cross-validate the TCSSE by applying it also to different groups of teachers (i.e., middle school teachers). Secondly, the samples present a high prevalence of females. These sample sizes over-represented by females are not surprising because the latest EU data available on teachers reported that, in Italy, 83.2% of teachers are female [49]. Unfortunately, the number of teachers within groups did not allow for the measurement invariance test [32,50,51]. Future investigation should explore the validity of the TCSSE across male and female groups. Thirdly, the survey consists of self-report instruments that may have been influenced by well-known biases, such as social desirability. Fourthly, the 4 week interval could be too short a time for a carryover effect and future studies could consider a longer interval of time to confirm the reliability of TCSSE. Finally, it must be acknowledged that the measure was developed and validated in the Italian language. The psychometric findings of the TCSSE could vary for teachers in different cultural contexts and/or different education systems. Hence, it would be appropriate to replicate our findings with other countries to test and extend the applicability of the TCSSE to their populations. Indeed, there is a need to consider the possibility that orientation-related activities that can be handled by teachers may vary according to different educational settings. 

Despite this limitation, the study contributes theoretically and practically to existing research in the career field. TCSSE is the first measure to assess teachers’ career-related support self-efficacy. We believe that the use of TCSSE could be fruitfully applied to the practice of school career counselors, allowing them to assess teacher perceived efficacy in providing career guidance and to test for the efficacy of specific training aimed at empowering teachers to support students’ career choices. Specifically, the TCSSE could be a starting point for developing specific training for teachers to help them develop increasingly effective career support for students.

## 6. Conclusions

Focusing on teacher support in preparing adolescents for career transitions is crucial because teachers have a key role in fostering them. Overall, results indicated that the TCSSE is a valid measure for assessing teacher career-related support self-efficacy. Findings support a six-factor structure of TCSSE—Get Ready, Empower Self, Get Curious, Empower Skills, Emotional Support and Instrumental Support. 

The good psychometric properties make the TCSSE a reliable measure for both educational and research settings. Therefore, career practitioners and researchers are encouraged to consider using the TCSSE in their practical and research activities.

## Figures and Tables

**Table 1 behavsci-13-00036-t001:** Item descriptive statistics and CFA.

	Descriptive Analysis				CFA	
	Mean	SD	Sk	K	λ	R^2^
Item 1	4.14	0.754	0.142	0.625	0.815	0.664
Item 2	4.22	0.723	0.142	−0.583	0.871	0.759
Item 3	4.06	0.748	0.142	−0.898	0.873	0.762
Item 4	4.33	0.674	0.142	−0.76	0.876	0.768
Item 5	4.22	0.739	0.142	−0.493	0.695	0.484
Item 6	4.19	0.807	0.142	0.779	0.731	0.535
Item 7	4.39	0.681	0.142	−0.316	0.876	0.767
Item 8	4.24	0.692	0.142	−0.888	0.832	0.691
Item 9	4.29	0.731	0.142	−0.523	0.805	0.649
Item 10	4.31	0.702	0.142	−0.87	0.869	0.755
Item 11	4.38	0.685	0.142	−0.692	0.865	0.748
Item 12	4.33	0.688	0.142	0.433	0.793	0.630
Item 13	4.26	0.72	0.142	−0.754	0.817	0.668
Item 14	4.1	0.786	0.142	−0.698	0.826	0.682
Item 15	4.21	0.726	0.142	−0.623	0.845	0.714
Item 16	4.34	0.665	0.142	−0.733	0.814	0.662
Item 17	4.33	0.664	0.142	−0.735	0.925	0.856
Item 18	4.33	0.689	0.142	−0.799	0.865	0.748
Item 19	4.38	0.675	0.142	−0.313	0.819	0.671
Item 20	4.39	0.701	0.142	−0.079	0.835	0.697
Item 21	4.32	0.721	0.142	−0.391	0.744	0.553
Item 22	4.37	0.693	0.142	−0.415	0.754	0.568
Item 23	4.23	0.772	0.142	0.239	0.754	0.568
Item 24	4.18	0.771	0.142	0.283	0.829	0.685
Item 25	4.28	0.68	0.142	−0.823	0.849	0.720
Item 26	4.07	0.84	−0.582	−0.163	0.735	0.540
Item 27	4.32	0.671	−0.548	−0.427	0.897	0.805
Item 28	4.39	0.666	−0.711	−0.263	0.938	0.880
Item 29	4.39	0.635	−0.552	−0.625	0.930	0.864
Item 30	4.39	0.64	−0.569	−0.625	0.859	0.738
Item 31	4.43	0.612	−0.565	−0.593	0.874	0.763
Item 32	4	0.865	−0.772	0.741	0.653	0.426
Item 33	4.02	0.801	−0.479	0.132	0.774	0.599
Item 34	4.23	0.693	−0.412	−0.632	0.907	0.823
Item 35	4.16	0.761	−0.605	0.179	0.885	0.784
Item 36	4.45	0.598	−0.574	−0.592	0.866	0.751

Note. SD = standard deviation; Sk = skewness; K = kurtosis; CFA = confirmatory factor analysis; λ = standardized factor loading into the specific factor (i.e., Get Ready, Empower Self, Get Curious, Empower Skills, Emotional Support, or Instrumental Support). All λ are statistically significant with *p* < 0.001. R^2^ = explained variance.

**Table 2 behavsci-13-00036-t002:** Correlation among latent factors.

	1	2	3	4	5	6
1. Get Ready	-					
2. Empower Self	0.432	-				
3. Get Curious	0.644	0.496	-			
4. Empower Skills	0.552	0.440	0.601	-		
5. Emotional Support	0.596	0.465	0.656	0.610	-	
6. Instrumental Support	0.566	0.461	0.641	0.697	0.665	-

Note. All correlations are statistically significant with *p* < 0.001.

**Table 3 behavsci-13-00036-t003:** Correlations between the TCSSE dimensions, career decision self-efficacy dimensions and teacher self-efficacy.

	1	2	3	4	5	6	7	8	9	10	11	12
1. Get Ready	-											
2. Empower Self	0.606	-										
3. Get Curious	0.801	0.657	-									
4. Empower Skills	0.726	0.634	0.758	-								
5. Emotional Support	0.758	0.646	0.794	0.794	-							
6. Instrumental Support	0.706	0.624	0.757	0.770	0.809	-						
7. CDSES_SE	0.492	0.450	0.569	0.513	0.568	0.547	-					
8. CDSES_OI	0.469	0.392	0.570	0.444	0.536	0.541	0.765	-				
9. CDSES_GS	0.490	0.425	0.537	0.480	0.545	0.534	0.845	0.811	-			
10. CDSES_PL	0.466	0.430	0.574	0.513	0.548	0.539	0.838	0.816	0.828	-		
11. CDSES_PS	0.485	0.416	0.547	0.481	0.526	0.540	0.821	0.761	0.766	0.829	-	
12. TSE	0.422	0.378	0.519	0.491	0.524	0.497	0.634	0.559	0.628	0.637	0.581	-

Note. CDSES = Career Decision Self-Efficacy Scale with SE = Self-appraisal, OI = Occupational information, GS = Goal selection, PL = Planning, PS = Problem Solving; TSE = Teacher Self-Efficacy. All correlations are statistically significant with *p* < 0.001.

## Data Availability

The data presented in this study are available on request from the corresponding author. The data are not publicly available due to privacy issues.

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
