# Peer review of "Development and Validation of the Teacher Career-Related Support Self-Efficacy (TCSSE) Questionnaire"

_behavsci, 2022, doi:10.3390/bs13010036_

Round 1

Reviewer 1 Report

ID: behavsci-2089250

Title: Development and Validation of the Teacher Career-related 2 Support Self-efficacy (TCSSE) questionnaire

Thank you for providing a chance to review this manuscript.

Comment: Major revision.

Detailed information:

Abstract:

Line 9, Page 1: Abstract needs to include the basic components of Background, Methods, Result, Conclusion, etc. Your abstract is not clearly chunked and multiple sections are mixed together. It is recommended to rewrite it.

Line 15, Page 1: You conducted three studies, what sample size was included in each study? and the main results of Study 1 are not accounted for.

Introduction

Line 34, Page 1: The argument in this paragraph seems disconnected from the first sentence. How does the list of challenging career paths justify to “Focusing on teacher support ...... increasingly crucial”? Please adjust the composition of the literature cited in this paragraph to make an effective argument.

Overall: In my opinion, the Introduction does too much padding about “Career-related teacher support” and seems very lengthy, so please simplify him.

Line 132, Page 3: What is your point of innovation? What are the implications of constructing such a scale to measure teacher career support efficacy? What direct positive impact can it have on students’ career choices? Please direct list it.

Study 1

Line 143, Page 3: The chunks in this section are confusing, can't Study1,2,3 be placed together under the Methods and Results chunk? Besides, “item generation through literature review……”, what are the main takeaways from the literature review? This is to be detailed in the Introduction section.

Line 182, Page 4: “29% were males and 71% were females”, the sample size was over-represented by females, could this lead to bias? How did you avoid these biases?

Study 2

Line 224, Page 5: How did you determine that the sample size was adequate? Were there no detailed inclusion and exclusion criteria when recruiting the sample size? This issue is present in all of Study1,2,3, please elaborate.

Line 272, Page 6: Do you have any descriptions of the characteristics of samples? We suggest adding a table about it.

Discussion

Overall: A general remark that I have is that the discussion lacks more in-depth conclusions. Instead, it is rich in data repeated from the Results sections.

Thank you and my best,

Your reviewer

Author Response

Abstract:

Line 9, Page 1: Abstract needs to include the basic components of Background, Methods, Result, Conclusion, etc. Your abstract is not clearly chunked and multiple sections are mixed together. It is recommended to rewrite it.

Line 15, Page 1: You conducted three studies, what sample size was included in each study? and the main results of Study 1 are not accounted for.

Thank you for this suggestion. We modified the abstract in accordance with the proposed structure (Background, Methods, Result, Conclusion) and fixed the contents. In addition, we added the sample size for each study and the main results of study 1.Changes are highlighted in yellow in the abstract.

Introduction

Line 34, Page 1: The argument in this paragraph seems disconnected from the first sentence. How does the list of challenging career paths justify to “Focusing on teacher support ...... increasingly crucial”? Please adjust the composition of the literature cited in this paragraph to make an effective argument.

Overall: In my opinion, the Introduction does too much padding about “Career-related teacher support” and seems very lengthy, so please simplify him.

We re-read the introduction to agree that the sentence seemed disconnected. We removed the sentence and revised the introduction to make it more coherent. Thank you!

Line 132, Page 3: What is your point of innovation? What are the implications of constructing such a scale to measure teacher career support efficacy? What direct positive impact can it have on students’ career choices? Please direct list it.

Thank you for this suggestion that allowed us to better outline the strengths of our work. We have outlined this more fully in the final part of the introduction.

Study 1

Line 143, Page 3: The chunks in this section are confusing, can't Study1,2,3 be placed together under the Methods and Results chunk? Besides, “item generation through literature review……”, what are the main takeaways from the literature review? This is to be detailed in the Introduction section.

Thank you for this suggestion. For the division into paragraphs, we followed the recommendations of Boateng and colleagues (2018). The literature review was specified in the introduction.

Line 182, Page 4: “29% were males and 71% were females”, the sample size was over-represented by females, could this lead to bias? How did you avoid these biases?

We know that the sample size was over-represented by females. For this reason, we have not carried out analysis of invariance between males and females. We addressed this problem in the limitation of the study. We also notify that there is great difficulty in finding male teachers because, as the latest EU data (2022) report, 83.2% of in-service teachers in Italy are female.

Study 2

Line 224, Page 5: How did you determine that the sample size was adequate? Were there no detailed inclusion and exclusion criteria when recruiting the sample size? This issue is present in all of Study1,2,3, please elaborate.

Thank you for this comment. Inclusion and inclusion criteria were added for the 3 studies.

Line 272, Page 6: Do you have any descriptions of the characteristics of samples? We suggest adding a table about it.

In addition to gender and age, we asked participants to indicate their years of experience as teachers. This information was reported in the paragraph of participants in each study.

Discussion

Overall: A general remark that I have is that the discussion lacks more in-depth conclusions. Instead, it is rich in data repeated from the Results sections.

We edited the discussion section unifying and deleting repetitions from the results sections and adding discussion on some aspects.

Thank you and my best,

Thank you again!

Reviewer 2 Report

Dear Authors,

 The research topic is relevant, although not new. One gets the impression that the novelty is aimed at connecting self-efficacy with career-related teacher support. In the theoretical part of the work, we miss the theoretical justification of the concept of self-efficacy, and even more so we miss the explanation of career-related teacher support self-efficacy. We notice that more attention is paid to career-related teacher support in the theoretical part of the work. We miss self-efficacy in the question wording of the questionnaire. The questionnaire gives the impression that career-related teacher support is being investigated.

We recommend improving the structure of the results section, it is complex and confusing. We doubt the necessity of Table 1. We suggest adding it to the appendix. We doubt the accuracy of the title of the third table, career decision self-efficacy or career-related teacher support self-efficacy? We suggest improving the layout of the third table.

We suggest improving the discussion section. The research methodology is repeated in the discussion. We missed the discussion in the discussion section. Descriptive information dominates in the discussion section. We are missing literature sources related to the theoretical constructs: self-efficacy and career-related teacher support self-efficacy. We noticed that data processing sources dominate the bibliography.

We suggest removing the concept Italian validation from the list of keywords or justifying it in the text of article.

I would like a lower percentage of coincidences (plagiarism). We suggest reducing plagiarism in the results section.

Author Response

Dear Authors,

 The research topic is relevant, although not new. One gets the impression that the novelty is aimed at connecting self-efficacy with career-related teacher support. In the theoretical part of the work, we miss the theoretical justification of the concept of self-efficacy, and even more so we miss the explanation of career-related teacher support self-efficacy. We notice that more attention is paid to career-related teacher support in the theoretical part of the work. We miss self-efficacy in the question wording of the questionnaire. The questionnaire gives the impression that career-related teacher support is being investigated.

Thank you for this suggestion that allowed us to better outline the strengths of our work. We have outlined this more fully in the final part of the introduction and focused the introduction in light of our topic. Changes are highlighted in yellow.

We recommend improving the structure of the results section, it is complex and confusing. We doubt the necessity of Table 1. We suggest adding it to the appendix. We doubt the accuracy of the title of the third table, career decision self-efficacy or career-related teacher support self-efficacy? We suggest improving the layout of the third table.

Thank you for the comment. Regarding Table 1, the Table contains the beta coefficients which are the main results of the CFA. However, we remain available to move the table to the appendix if the journal deems it necessary for editing issues. For the table 3 title, TCSSE stands for career-related teacher support self-efficacy while career decision self-efficacy refers to the Career Decision Self-Efficacy Scale questionnaire (Lo Presti et al., 2013; Taylor & Betz, 1983) used to test convergent validity. We have included Table 3 (instead of the image) improving the layout but, again, we are available for other editing changes in line with journal standard.

We suggest improving the discussion section. The research methodology is repeated in the discussion. We missed the discussion in the discussion section. Descriptive information dominates in the discussion section. We are missing literature sources related to the theoretical constructs: self-efficacy and career-related teacher support self-efficacy. We noticed that data processing sources dominate the bibliography.

We edited the discussion section unifying and deleting repetitions from the results sections and adding discussion on some aspects. Thank you for your suggestions.

We suggest removing the concept Italian validation from the list of keywords or justifying it in the text of article.

Done! Thank you!

I would like a lower percentage of coincidences (plagiarism). We suggest reducing plagiarism in the results section.

We checked the results. Thank you!

Round 2

Reviewer 1 Report

ID: behavsci-2089250

Title: Development and Validation of the Teacher Career-related Support Self-efficacy (TCSSE) questionnaire

Thank you for providing a chance to review this manuscript.

Comment: minor revision.

Detailed information:

Abstract

       Line 9, Page 1: I suggest you streamline the Background section, some of the descriptions should be placed in Introduction , eg the definition of “career-related teacher support”.

Line 18, Page 1: I suggest you list some core statistical values in the Methods section, eg alpha value.

Study 2

Line 243~261, Page 5~6: Have these two scales been validated in the region? Please add psychometric properties of both the original scales and the Italian version scales.

Conclusion

       Please add a Conclusion section to summarise the main findings of this study in plain language.

Table

       Table 1If Table 1 spans pages, you can put the corresponding table header on it to make it easier for readers to read.

       The problems mentioned before have been improved, congratulations! Besides, some minor but important issues remain, You need to revise these issues to make the article better.

Thank you and my best,

Your reviewer

Author Response

Dear Reviewer,

thank you for your comments that allowed us to improve the quality of our paper. We have taken into account the suggestions made in this second round of review. Below is our point-by-point response.

Thank you again!

Detailed information:

Abstract

       Line 9, Page 1: I suggest you streamline the Background section, some of the descriptions should be placed in Introduction , eg the definition of “career-related teacher support”.

Line 18, Page 1: I suggest you list some core statistical values in the Methods section, eg alpha value.

We modified the abstract following suggestions. Highlighted in green you can finds new revisions.

Study 2

Line 243~261, Page 5~6: Have these two scales been validated in the region? Please add psychometric properties of both the original scales and the Italian version scales.

Done! We have added the alphas of original and Italian validation measures.

Conclusion

       Please add a Conclusion section to summarise the main findings of this study in plain language.

 Done! We have added the conclusion. Thank you.

Table

       Table 1If Table 1 spans pages, you can put the corresponding table header on it to make it easier for readers to read.

 Thank you for this valuable suggestion. We do not know if this is allowed by the journal, but we will ask for this possibility at the proofreading process.

       The problems mentioned before have been improved, congratulations! Besides, some minor but important issues remain, You need to revise these issues to make the article better.

Thank you!

Reviewer 2 Report

The definition of Career-related teacher support self-efficacy (P3_ 110) is questionable. Career-related teacher support self-efficacy??? can be defined as the teachers' perceived self-efficacy??? in different tasks related to career support.

We propose to improve the definition of Career-related teacher support self-efficacy. Career-related teacher support self-efficacy can be defined as the teachers' perceived estimated probability of success in a different task related to students' career support.

Author Response

Thank you for this remark. We have modified the definition of Career-related teacher support self-efficacy following this suggestion.